# Role of Whole Grain Consumption in Glycaemic Control of Diabetic Patients: A Systematic Review and Meta-Analysis of Randomized Controlled Trials

**DOI:** 10.3390/nu14010109

**Published:** 2021-12-27

**Authors:** Dengfeng Xu, Lingmeng Fu, Da Pan, Yifei Lu, Chao Yang, Yuanyuan Wang, Shaokang Wang, Guiju Sun

**Affiliations:** 1Key Laboratory of Environmental Medicine and Engineering of Ministry of Education, Department of Nutrition and Food Hygiene, School of Public Health, Southeast University, Nanjing 210009, China; withxu@seu.edu.cn (D.X.); dapan@seu.edu.cn (D.P.); 230198332@seu.edu.cn (Y.L.); 230189316@seu.edu.cn (C.Y.); wyy@seu.edu.cn (Y.W.); shaokangwang@seu.edu.cn (S.W.); 2Department of Quality Management, Zhejiang Provincial People’s Hospital, Hangzhou 310014, China; 220173455@seu.edu.cn

**Keywords:** whole grains, diabetes mellitus, meta-analysis, randomized controlled trials

## Abstract

Background: Observational studies have indicated beneficial effects of whole grain consumption on human health. However, no evidence based on randomized controlled trials has been established. Our objective was to perform a systematic review and meta-analysis of randomized controlled trials to assess the effects of whole grain consumption in glycaemic control of diabetic patients. Methods: A comprehensive search in four databases (Web of Science, Pubmed, Scopus and Cochrane library) was conducted to collect potential articles which measured the roles of whole grain consumption on glycaemic control up to October 2021. Results: A total of 16 eligible trials involving 1068 subjects were identified to evaluate the pooled effect. The overall results indicated that compared with the control group, whole grain intake presented a significantly reduced concentration in fast plasma glucose (WMD = −0.51 mmol/L, 95% CI: −0.73, −0.28; $I^{2}$ = 88.6%, *p* < 0.001), a homeostasis model assessment of insulin resistance (WMD = −0.39 μU × mol/L^2^, 95% CI: −0.73, −0.04; $I^{2}$ = 58.4%, *p* = 0.014), and glycosylated haemoglobin (WMD = −0.56%, 95% CI: −0.88, −0.25, $I^{2}$ = 88.5%, *p* < 0.001), while no significant difference was observed in fast plasma insulin level between groups (SMD = −0.05, 95% CI: −0.25, 0.14; $I^{2}$ = 40.7%, *p* = 0.120). In terms of incremental area under the curve (iAUC), data suggested that whole grain effected a significant decrease in Glucose-iAUC (WMD = −233.09 min × mmol/L, 95% CI: −451.62, −14.57; $I^{2}$ = 96.1%, *p* < 0.001) and Insulin-iAUC (SMD = −4.80, 95% CI: −8.36, −1.23; $I^{2}$ = 89.9%, *p* = 0.002), although only in a small number of studies. Of note, there is evidence for modest unexplained heterogeneity in the present meta-analysis. Conclusion: Whole grain consumption confers a beneficial effect on glucose metabolism in patients with diabetes. Regrettably, since relevant studies were scarce, we failed to provide confident evidence of whole grain consumption on acute effects including Glucose-iAUC and Insulin-iAUC, which should be addressed in further trials.

## 1. Introduction

Diabetes mellitus (DM), which is usually characterized by an absolute or relative deficiency of serum insulin concentrations, is one of most serious metabolic diseases with a sharply increasing incidence globally. It is estimated that there were about 537 million diabetics among the population aged from 20–79 worldwide in 2021, and this number is expected to reach 784 million in 2045 [1]. According to a report from the World Health Organization, since 1980, the global prevalence of DM has almost doubled from 4.7% to 8.5% in the adult population [2]. In China, there is one diabetic patient in each 12 adults; worse, the estimated prevalence of prediabetes had risen to 35.7% in 2017 [3], indicating a serious economic burden in the future.

DM is caused by both genetic and environmental factors. Findings from mechanistic studies suggest that DM can lead to abnormal changes including metabolic profiles, energy production, redox status, and extracellular matrix remodelling [4], eventually resulting in atherosclerotic cardiovascular diseases. It is reported that adults with diabetes have a 2–4 times higher cardiovascular risk than adults without diabetes [5]. A meta-analysis performed by Einarson et al. indicated that about half of the deaths in patients with type 2 diabetes mellitus (T2DM) comprise cardiovascular diseases (CVDs) [6]. Therefore, we believe that poor management of DM patients can increase the possibility of suffering from reduced life quality and expectancy.

Previous studies have provided credible evidence for the key role of diet in the prevention of DM [7,8]. Whole grains are a group of cereal foods in which the endosperm, germ, and bran are intact. They are also a good source of dietary fibres, vitamins, antioxidants, and phytochemicals [9], such as phenolic compounds (including ferulic acids and cinnamic), beta-glucan, and lignans, which have been reported to play a protective role in many metabolic diseases, such as T2DM, obesity and CVDs [10,11]. The gut microbiota is referred to as a community of more than 10^14^ bacteria residing in the human intestine, and emerging discoveries have suggested that dysbiosis of the gut microbiota can result in many age-related and lifestyle diseases such as diabetes, obesity, inflammation, and CVDs [12,13,14]. A study by Zhao et al. suggested that T2DM could be alleviated by dietary fibres, which can be fermented by the gut microbiota and produce the beneficial metabolites of short-chain fatty acids (SCFAs) [15]. Indeed, a 6-week randomized controlled parallel-design trial conducted in postmenopausal women indicted that whole grain consumption led to an increased abundance of *Lachnospira* and the level of stool acetate; furthermore, *Lachnospira* was positively associated with acetate [16]. Similarly, a cross-sectional study has proposed significant associations between circulating SCFA, particularly acetate and propionate, and peripheral insulin sensitivity, whole body lipolysis and glucagon-like peptide-1 (GLP-1) concentrations [17]. In an in vivo study, Lappi et al. and Nilsson et al. also found that the consumption of barley kernel-based bread or wholegrain rye bread resulted in improved markers of glucose metabolism with increased serum SCFAs concentrations in healthy individuals [18,19]. A meta-analysis of cohort studies has established an inverse association between whole grain intake and the risk of T2DM [20]. However, the results are inconsistent in clinical trials. Li et al. found that replacing a healthy diet (low-fat and high-fibre diet) with 100 g of oats for 30 days leaded to a significant improvement in fast plasma glucose (FPG), glycosylated hemoglobin (HbA1c), and homeostasis model assessment of insulin resistance (HOMA-IR) in overweight diabetics [21]. On the other hand, a study by McGeoch et al. showed no significant differences in terms of glycaemic control following intervention with the oat-enriched diet [22]. This discrepancy may be a consequence of differences in population selection and the type and amount of whole grain production consumed. Given the inconsistent results from clinical randomized controlled trials and the absence of any systematic reviews and meta-analyses, it is necessary to perform the present study.

Accordingly, the purpose of this study was to perform a systematic review and meta-analysis of randomized controlled trials to evaluate the role of whole grain consumption in glycaemic control in diabetic patients and make some suggestions regarding diabetes diets.

## 2. Methods

### 2.1. Search Strategy and Study Selection

Eligible studies were identified by a comprehensive search up to October 2021 from the Web of Science, Pubmed, Scopus, and Cochrane library databases. The languages were limited to English only. The search was implemented based on a combination of Mesh terms and keywords as follows: “whole grains or whole-grains or grains or cereals or whole wheat or barley or oat and diabetes mellitus”. The detailed search strategies and results are shown in the Appendix A. The references in the papers were reviewed as well so that any relevant studies were not missed.

Studies were selected if they met the following criteria: (i) study design must be a randomized controlled trial which evaluate the effects of whole grains on glycaemic control; (ii) study was performed on diabetics or prediabetics; (iii) inclusion of an appropriate control group in which a diet had low or no whole grain intake; (iv) reported data with available mean change from endpoint to baseline and any of standard deviation (SD), standard error or 95% confidence interval for the interested outcomes. The exclusion criteria were the following: (i) research subjects were animals or cells; (ii) study assessed only individual components of grain, such as dietary fibres; (iii) intervention diets were combined with other meals which may confuse the effects of whole grain; (iv) secondary information, such as reviews.

### 2.2. Data Extraction and Study Quality

The literature information was extracted independently by two authors (Xu and Fu), any discrepancies were resolved by discussion with a third author (Pan). The detailed information collected from each eligible study included: first author, year (country), sample size (male/female), average ages of participants, intervention meals, amounts of whole grains, matching meals, type of diabetes mellitus, outcomes evaluated, duration, and study design. In addition, net mean changes in FPG, fast plasma insulin (FPI), HOMA-IR, HbA1c (%), glucose-incremental area under the curve (G-iAUC), and insulin-incremental area under the curve (I-iAUC) in each group were collected through the following formula:$${\mathrm{Mean}}_{\mathrm{net}\;\mathrm{changes}}={\;\mathrm{Mean}}_{\mathrm{endpoint}\;}-{\mathrm{Mean}}_{\mathrm{baseline}\;}$$

For the calculation of net changes of SD, following formula was used:$${\mathrm{SD}}_{\mathrm{net}\;\mathrm{change}}=\sqrt{\mathrm{SD}{\left(\mathrm{baseline}\right)}^{2}+\mathrm{SD}{\left(\mathrm{endpoint}\right)}^{2}-2R\times \mathrm{SD}\left(\mathrm{baseline}\right)\times \mathrm{SD}\left(\mathrm{endpoint}\right)}$$

The correlation coefficient R can be designated as 0.5 according to Higgins [23]. Moreover, when the interested outcomes (including FPG and G-iAUC) were reported with units of mg/dL and min × mg/dL, we converted them to the mmol/L and min × mmol/L, respectively (multiply mg/dL by 0.0555) and evaluated them with weight mean difference (WMD); for FPI and I-iAUC, standardized mean difference (SMD) was used to assess the pooled effect.

Quality assessment was assessed by the Cochrane Risk of Bias Tool [23], which mainly covers seven validity questions: random sequence generation, allocation concealment, blinding of participants and personnel, blinding of outcome assessment, incomplete outcome data, selective reporting, and other sources of bias; each item was scored as high risk (if it contained methodological flaws that may have affected the results), unclear risk (if information was insufficient to determine the impact), or low risk (if the flaw was deemed inconsequential) of bias for all included studies. We considered a study as low quality if that study was scored as high risk for at least 2 domains of the Cochrane Risk of Bias Tool; if a study was labelled as low risk for all domains, it was considered as a high-quality study. Upon a critical appraisal by one author (Wang) for each article, a table of overall risk of bias was derived.

### 2.3. Statistical Methods

The $I^{2}$ statistic (range from 0–100%) was employed to assess the heterogeneity between studies, and when $I^{2}$ is greater than 50%, it was considered to indicate a high level of heterogeneity and the random-effects model was used; otherwise, a fixed-effects model was used [24]. Funnel plots and Egger’s regression test were created to explore potential publication bias, and ‘trim and fill’ analysis was used to further observe the stability of results if there is any asymmetry in the funnel plot [25]. In addition, we performed a sensitivity analysis to explore the sources of heterogeneity by removing each study and then repeating the analysis. Subgroup analysis was carried out to evaluate some potential factors, including study design, type of diabetes mellitus, duration of the trials, and type of controlled diet. All analyses were conducted in Stata version 12, and *p <* 0.05 was regarded as statistically significant.

## 3. Results

### 3.1. Literature Selection

Based on the established search strategy, a total of 7832 studies were initially assessed, in which 243 duplications and 7465 irrelevant studies were excluded after title and abstract review, leading to a total of 124 articles for further assessment. After reading the full text of the remaining articles, 109 articles were removed. The reasons were as follows: (i) no whole grains (n = 58); (ii) without interesting outcomes (n = 21); (iii) with inappropriate control (n = 8); (iv) reviews or meeting abstract (n = 22). Finally, 16 studies (including one study added from the references) were eligible for the present quantitative meta-analysis. The detailed flow diagram for the selected trials is shown in Figure 1.

### 3.2. Study Characteristics

Table 1 summarized the characteristics of the 16 eligible studies, which included 1068 diabetic or prediabetic subjects (11 studies were performed on patients with T2DM, 1 on patients with gestational diabetes, and the other 4 were on prediabetics). There were nine studies conducted in developed countries (New Zealand, USA, Germany, Japan, Finland, and Canada), while the other seven studies were performed in developing countries (Iran, India, China, and Bangladesh). The average age of participants ranged from 28.72 year to 68.10 year, with a ratio of male to female of 46.5% to 53.5%. The intervention meals involved oat bran, low GI whole grain, black-grained wheat, brown rice, barley flakes, and others. Correspondingly, the matching foods included rice, refined whole grain, usual diet, systematic diet plans and others. The intervention duration of included articles varies from 4 weeks to 1 year, and all studies were performed by randomized controlled trials, and half of them are crossover and half of them are parallel design.

Additionally, based on the criteria of the Cochrane Risk of Bias Tool, a risk of bias for the quality of the included studies is shown in the Appendix A.

### 3.3. Effects of Whole Grain Consumption on Parameters Involve Glycaemic Control

Since about 1980, because of shifts in dietary patterns and a more sedentary lifestyle, the global prevalence of DM has nearly doubled. Raised blood glucose, a common effect of uncontrolled diabetes, may, over time, lead to serious damage to the heart, blood vessels, eyes, kidneys, and nerves. However, results from a meta-analysis of prospective studies indicated that whole grain may have beneficial effects on glucose modulation [8]. These findings provide us with a new strategy for managing DM and avoiding undesirable effects caused by drug treatments.

#### 3.3.1. Fast Plasma GLUCOSE Concentrations

The pooled effects of whole grain consumption on FPG concentrations were reported by fourteen studies in twelve articles, which showed a significant decrease in FPG concentrations (WMD = −0.51 mmol/L, 95% CI: −0.73, −0.28, Figure 2A) in subjects who consumed whole grain than those in a control group, while substantial evidence of heterogeneity was found between the studies ($I^{2}$ = 88.6%, *p* < 0.001, Figure 2A). Of note, the results of subgroup analysis by national economic level indicated no heterogeneity within studies performed in developed countries; moreover, no significant difference was observed between groups (WMD = −0.03 mmol/L, 95% CI: −0.11, 0.04, Table 2).

#### 3.3.2. Fast Plasma Insulin Concentrations

The present meta-analysis consisting of seven studies demonstrated that, compared with the control group, the concentrations of FPI were not significantly changed by whole grain consumption (SMD = −0.05, 95% CI: −0.25, 0.14, Figure 2B), with a moderate heterogeneity ($I^{2}$ = 40.7%, *p* = 0.120, Figure 2B). In addition, we found a lower heterogeneity when it comes to the subgroup analysis by matching foods (0% for standard diet and 34.7% for others), which suggested that the controlled foods may play an important role in the evaluated outcomes. More detailed subgroup analysis is summarized in Table 2.

#### 3.3.3. Homeostasis Model Assessment of Insulin Resistance

Data on the effects of whole grain consumption on HOMA-IR were calculated by nine studies based on seven articles. As is shown in Figure 2, whole grain consumption led to a significant reduction in HOMA-IR (WMD = −0.39 μU × mol/L^2^, 95% CI: −0.73, −0.04, Figure 2C), though with obvious heterogeneity ($I^{2}$ = 58.4%, *p* = 0.014, Figure 2C). Further subgroup analysis suggested that the pooled effect concluded from three studies conducted in developed countries was not significant and with low heterogeneity, which showed some similarities to FPG. The remaining results of the subgroup analysis can be found in Table 2.

#### 3.3.4. Glycosylated Haemoglobin

Overall, nine effect sizes from seven clinical trials were eligible to assess the pooled effects of whole grain consumption on HbA1c in diabetic patients. A notable decrease in pooled effect on HbA1c was observed after whole grain consumption (WMD = −0.56%, 95% CI: −0.88, −0.25, Figure 2D); however, there was still a serious heterogeneity before subgroup analysis ($I^{2}$ = 88.5%, *p* < 0.001, Figure 2D). Consequently, subgroup analysis was conducted to explore the potential sources of heterogeneity. Interestingly, when we focused on short-term studies (<8 weeks) or those conducted in developed countries, the results became non-significant. We speculated that the small number of studies of the subgroup analysis may be partly responsible for this situation. Table 2 showed the remaining results of the subgroup analysis.

#### 3.3.5. Glucose/Insulin Incremental Area under the Curve

Fewer studies had reported the effects of whole grain consumption on the G-iAUC and I-iAUC. As shown in Figure 2, the pooled WMD for whole grain on the G-iAUC was −233.09 min × mmol/L (95% CI: −451.62, −14.57; Figure 2E) and with high heterogeneity ($I^{2}$ = 96.1%, *p* < 0.001, Figure 2E); on the other hand, there were only two studies that reported data on I-iAUC, and the pooled SMD value was −4.80 (95% CI: −8.36, −1.23; Figure 2F). A serious heterogeneity ($I^{2}$ = 89.9%, *p* = 0.002, Figure 2F) was observed as well. Since the relevant studies were scarce, it is unnecessary to conduct any further subgroup analysis

### 3.4. Sensitivity Analysis

Sensitivity analysis, which acts as a method through sequentially removing each selected study and then repeating the meta-analysis, was employed for the purpose of a credible and stable pooled effects of whole grain intake on glycaemic control. The result showed that there is no significant change in the exclusion of any individual study (Appendix A).

### 3.5. Subgroup Analysis

The detailed pooled effects of whole grain intake on glycaemic control in subgroups based on subjects’ characteristics are presented in Table 2. Briefly, the subgroup analysis of FPG indicated that the overall pooled effects were not affected by the subgroups, including type of diabetes, matching foods, and durations. In terms of FPI, there is no significant difference from the overall effects even after adjusting all the subgroup factors. However, we only observed a significant reduction in HOMA-IR in the subgroup of subjects who had T2DM, or when the intervention duration is less than 8 weeks. Similarly, the overall effects of whole grain consumption on glycaemic control were also influenced by the subgroup of study design, matching foods, and intervention duration. Interestingly, when it comes to the studies performed in developed countries, all effects of whole grain consumption on glycaemic control became non-significant, although with a small sample size (Table 2).

### 3.6. Publication Bias

To explore whether there are potential publication biases, funnel plots and Egger’s regression tests were performed. The results from funnel plots of whole grain consumption on FPI, HbA1c, and G-iAUC showed good asymmetry, indicating that no bias was found; this was also confirmed by the *p* value of Egger’s regression of 0.79, 0.063, and 0.114, respectively (Figure 3 and Figure 4). However, there was minor asymmetry for FPG (Egger’s regression *p* = 0.005) and HOMA-IR (Egger’s regression *p* = 0.042); consequently, we further conducted the “trim and fill” method to assess the robustness of the results in the presence of publication bias, and the results were kept the same before and after adding the estimated missing literature (Appendix A), which indicating that the results of FPG and HOMA-IR observed in the present meta-analysis are reliable. Since the relevant article evaluating the I-iAUC was scarce, it is impossible to perform Egger’s regression test (Figure 4).

## 4. Discussion

To the best of our knowledge, the present study is the first systematic review and meta-analysis of randomized controlled trials to provide evidence for the effects of whole grain consumption on glucose metabolism in diabetic patients. Our results, concluded from 16 studies involving 1068 subjects, indicated that compared with the control group, whole grain consumption could significantly decrease the concentration of FPG, HOMA-IR, and HbA1c, while no significant difference was observed in FPI. Additionally, significant decreases in G-iAUC and I-iAUC after whole grain intake were also observed, although the relevant studies were scarce. Different matching foods and national economic levels may explain some unknown heterogeneity.

Recently, the results from a meta-analysis of prospective cohort studies suggested that compared with a low daily intake of dietary fibre, a 35 g daily intake could make an absolute reduction of 14 fewer deaths for each 1000 participants [41]. Indeed, a pilot study conducted by Khalil et al. suggested that a 12-week whole grain plant-based diet significantly decreased the level of FPG and HbA1c in newly diagnosed diabetics [42]. Consistently, our results also showed notable improvements in FPG and HbA1c after whole grain intake, in which HbA1c is more commonly used to diagnose and identify those at higher risk of developing diabetes in the future [43]. One proposed mechanism for those effects is that fibres contained in whole grain could increase the viscosity of intestinal content, which results in slowing down the absorption of glucose and eventually delaying the gastric emptying rate [44]. In an in vitro study, Abbasi et al. found that oat-derived β-glucan could significantly reduce the uptake of glucose in non-transformed rat small intestine epithelial cells by downregulating the expressions of glucose transporters sodium–glucose-linked transport protein 1 and glucose transporter 2 [45]. Moreover, the evidence for the relationship between the gut microbiota and health has been well-established, which drives us to consider the extensive role of the gut microbiota in the glucose metabolism. A study by Qin et al. indicated that the gut microbiota of patients with T2DM is mainly characterized by a decreased abundance of butyrate-producing bacteria, namely *Roseburia* and *Faecalibacterium prauznitzii,* compared with healthy subjects [46]. Furthermore, the gut microbiota was reported to be able to deconjugate bile acid, splitting off taurine or glycine molecules. Subsequently, the unconjugated bile acids can be dehydroxylated by specific strains from the *Clostridium* genus in the large intestine, which can act as better ligands for the farnesoid X receptor (FXR) and G protein-coupled (such as TGR5) receptors [47]. TGR5 activation in pancreatic α-cells induces pro-convertase-1 expression, shifting glucagon production to GLP-1, hence increasing β-cell mass and function in a paracrine manner [48], while FXR can reduce postprandial glucose utilization by inhibiting hepatic glycolysis and lipogenesis [49]. Interestingly, when we focused on the trials performed in developed countries, the beneficial effects of whole grain consumption on glucose metabolism became non-significant. Similarly, a randomized controlled trial conducted in Japan indicated that there is no effect of brown rice intake on glycolipid index, but not for parameters involving endothelial function [32]. On the other hand, we also found evidence supporting the beneficial effects of whole grains in subjects with metabolic diseases from a published meta-analysis [50]. Given that the study conducted in Japan did not assess the real increases in fibre intake in an experimental group; the number of local economic level-based studies in the present subgroup analysis is also relatively small. These biases may confound our results and need to be verified with well-designed trials in the future.

Insulin, which acts as a unique hormone that lowers the blood glucose in the body, plays a key role in maintaining the nutrient homeostasis during the postprandial state. It has been reported that insulin can exert many effects in a variety of tissues, including stimulating the influx of glucose into skeletal muscles and promoting the glycogen synthesis; on the other hand, in the liver and adipose tissues, insulin can suppress the production of hepatic glucose, and promote the synthesis and storage of lipids [51]. Indeed, a significant decrease in HOMA-IR value was observed upon whole grain consumption in our study. Similarly, according to Wirstrom et al., whole grain intake showed a negative correlation with HOMA-IR [52]. Moreover, evidence from a high-fat fed-diet animal study suggested that barley β-glucan could improve insulin sensitivity by decreasing serine phosphorylation of insulin receptor substrate 1 and activating Akt and downregulating the mRNA levels of glucose-6-phosphatase and phosphoenolpyruvate carboxykinase [53]. The possible mechanism behind this may be partly attributed to SCFAs, which are produced by bacterial fermentation for dietary fibres. It has been reported that the metabolic actions of SCFAs in improvements of insulin sensitivity including increasing glucose oxidation and insulin clearance and decreasing fatty acid release [54]. Moreover, other metabolites (including lipopolysaccharide, branched-chain amino acid and bile acids) from the gut microbiota may also play an important role in insulin resistance [55]. For example, Faits et al. found that consuming an unrefined carbohydrate diet resulted in a significant increase in abundance of *Roseburia*, as well as a decreased concentration of secondary bile acid [56]. Results from another randomized controlled crossover feeding study suggested that whole grain consumption led to significant increases in circulating concentrations of taurolithocholic acid, taurocholic acid, and glycocholic acid; moreover, significant associations between bile acids and HOMA-IR were also found in this study [57]. However, the present meta-analysis indicated that whole grain consumption could not improve FPI concentration significantly, even though various subgroup analyses were further conducted. This result is consistent with the meta-analysis performed by Marventano et al., in which they also found that compared with refined foods, the consumption of whole grain is not able to significantly decrease FPI levels in healthy subjects [58]. Of note, when we focus on the specific whole grain (e.g., oats), relevant evidence is still needed to be confirmed. Bao et al. suggested that oat intake resulted in a significant reduction in fasting insulin by −6.29 pmol/L [59], while in a meta-analysis performed by Shen et al., there is no significant effect of oat β-glucan intake on FPI, although only two studies were included [60]. Since the diet selections of the control group may vary when studies were performed with different objectives, in our opinion, the types of control diet are very important and should be taken into consideration when exploring the potential biases.

On the aspects of acute parameters, our results also showed significant decreases in G-iAUC and I-iAUC after whole grain consumption. Consistently, a meta-analysis performed in healthy participants also showed significant reductions in the postprandial values of the G-iAUC and I-iAUC after whole intake [58]. However, it is worth noting that the results concluded from the present study were based on a small sample size of involved studies and with a high heterogeneity. In a randomized crossover design trial conducted by Tucker et al., they claimed that whole grain sourdough bread could not significantly affect postprandial glucose and insulin in adults with T2DM [61]. We speculated that the discrepancy between studies may be related to differences in the selection of the population and control diets. In addition, due to the lack of relevant studies, it is impossible to conduct subgroup analysis, and the results should be explained with caution.

Compared with previous studies, our study has the following strengths: diabetes-related metabolic parameters, including FPG, insulin, HbA1c, HOMA-IR, as well as acute parameters such as G-iAUC and I-iAUC were all in our considerations. In addition, when we explored the sources of heterogeneity, we focused on the factors of matching foods and national economic levels, which may be ignored by others but may play an important role in the interpretation of inconsistent results. However, there are also some limitations that should be taken into consideration. Firstly, due to the lack of a well-accepted definition of whole grain, the selection of control diets became quite important. Inconsistent control diets may lead to some biases in our results. In addition, significant asymmetries in funnel plots for FPG and HOMA-IR will still be observed even though we had conducted a comprehensive search. However, further “trim and fill” analysis has delivered evidence for the robustness of the results. Additionally, even though we divided the national economic levels into developed and undeveloped during subgroup analysis, the real socioeconomic status of the participants may not be consistent with their national economic level. Finally, the number of studies included in the present meta-analysis is relatively small, which may also affect our results and need to be verified with more well-designed trials.

## 5. Conclusions

In summary, despite the significant reductions in whole grain consumption on G-iAUC and I-iAUC, we still recommend that the result should be considered with caution. Because the involved sample sizes in the present study are relatively small, additional high-quality RCTs conducted in human subjects with a parallel design are required to further investigate the effect of whole grain intake on those acute parameters. However, whole grain consumption has significant beneficial effects in medium- and long-term parameters, including decreasing concentrations of FPG, FPI, and HOMA-IR, which may provide evidence for the therapeutic potential of whole grain in diabetic patients or for preventing glucose dysregulation in those at a risk of DM.

## Figures and Tables

**Figure 1 nutrients-14-00109-f001:**
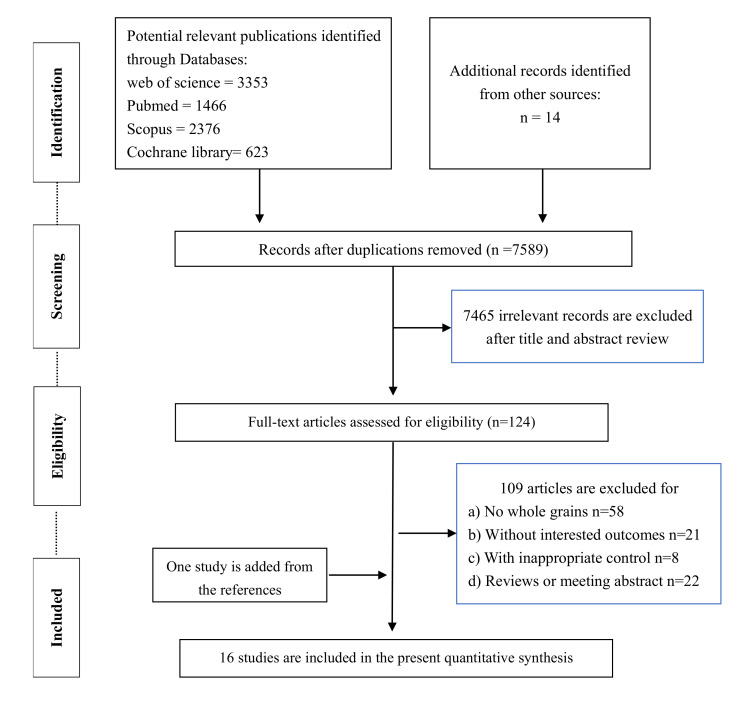
The flow diagram for selected trials.

**Figure 2 nutrients-14-00109-f002:**
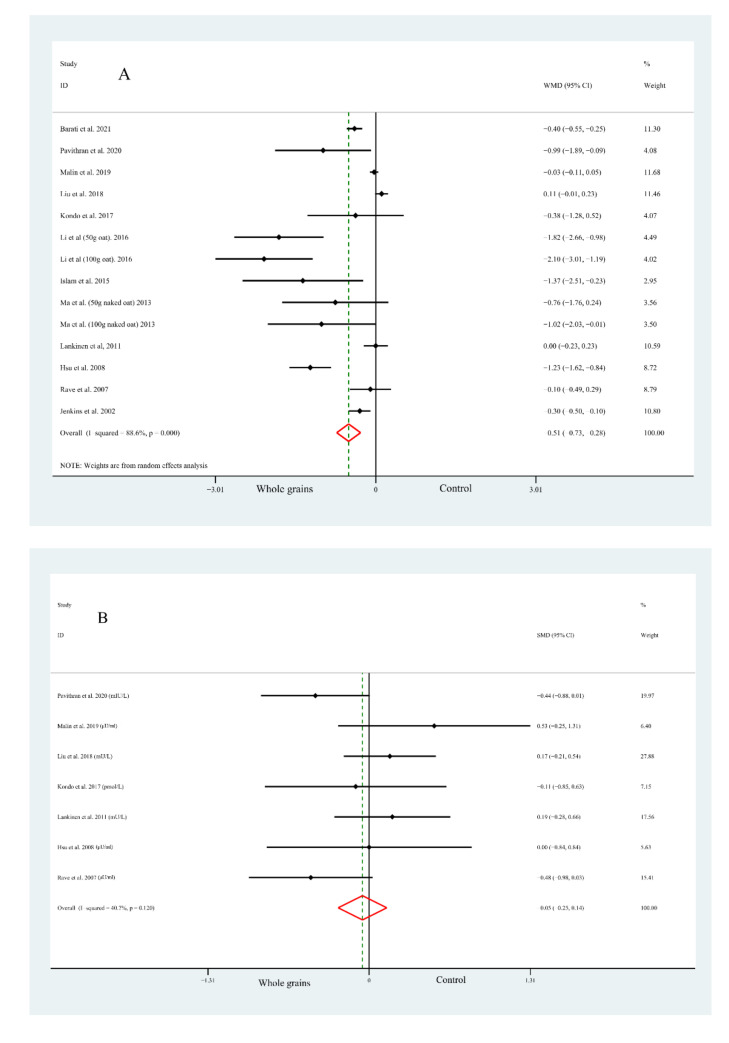
Forest plot of the effects of whole grain consumption on FPG (**A**), FPI (**B**), HOMA-IR (**C**), HbA1c (**D**), G-iAUC (**E**) and I-iAUC (**F**) in diabetic patients.

**Figure 3 nutrients-14-00109-f003:**
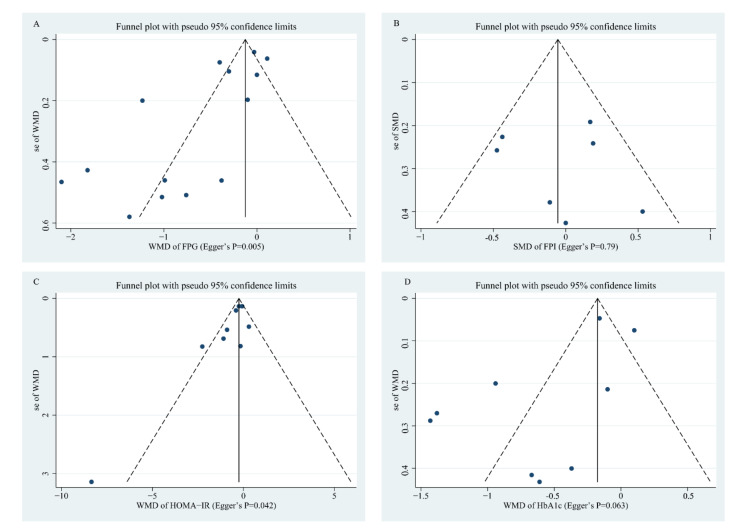
Funnel plot measuring the publication bias for the effects of whole grain consumption on FPG (**A**), FPI (**B**), HOMA-IR (**C**), and HbA1c (**D**) in diabetic patients.

**Figure 4 nutrients-14-00109-f004:**
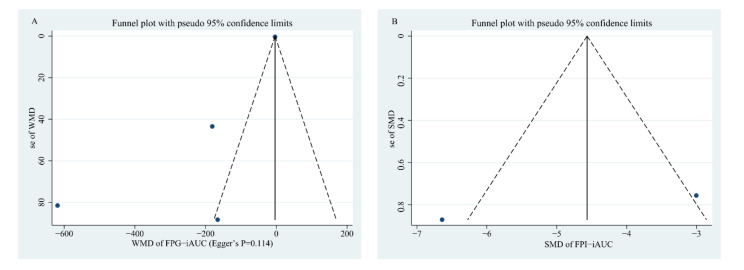
Funnel plot measuring the publication bias for the effects of whole grain consumption on G-iAUC (**A**) and I-iAUC (**B**) in diabetic patients.

**Table 1 nutrients-14-00109-t001:** Characteristics of included 16 trials for present meta-analysis.

First Author, Year (Country)	Sample Size (Male/ Female)	Age (Year)	Intervention Meals	Amounts of Whole Grains	Matching Foods	Type of Diabetes Mellitus	Outcomes Evaluated	Duration (Week)	Study Design
Elbalshy et al., 2021[26] (New Zealand)	18 (11/7)	63.10 ± 9.80	I1: Hot cooked Coarse kibbled whole grain I2: Cold uncooked Coarse kibbled whole grain	50 g	Finely milled whole grain	T2DM	G-iAUC	None	Randomized crossover trial
Barati et al., 2021[27] (Iran)	104 (0/104)	C = 28.72 ± 4.13; I = 29.23 ± 3.80	Standard diet with oat bran	30 g	Standard diet without oat bran	Gestational diabetes	FPG	4 weeks	Randomized controlled trial
Pavithran et al., 2020 [28] (India)	80 (52/28)	C = 51.93 ± 7.43; I = 54.43 ± 7.57	Local low GI whole grain	None	Usual diet	T2DM	FPG; FPI; HOMA-IR; HbA1c	24 weeks	Randomized controlled trial
Malin et al., 2019 [29] (USA)	13 (10/3)	37.20 ± 1.80	Whole grain	100	Refined-grain diet	Prediabetic adults	FPG; FPI;	8 weeks	Randomized controlled crossover trial
Delgado et al., 2019 [30] (Germany)	15 (8/7)	58.6 (10.1)	Oatmeal	100	Usual diet	T2DM	HbA1c	12 weeks	Randomized controlled crossover trial
Liu et al., 2018 [31] (China)	110 (50/60)	C = 57.40 ± 8.80; I = 58.50 ± 10.20	Black-grained wheat	50	Rice	T2DM	FPG; HbA1c; FPI; HOMA-IR	5-weeks	Randomized controlled trial
Kondo et al., 2017 [32] (Japan)	28 (18/10)	C = 68.10 ± 6.80; I = 65.20 ± 8.70	Brown rice	28–30 kcal/kg	White rice	T2DM	FPG; FPI; HbA1c; HOMA-IR	8 weeks	Randomized controlled trial
Li et al., 2016 [21] (China)	219 (113/106)	C = 59.00 ± 3.94; I1 = 59.72 ± 6.10; I2 = 59.44 ± 6.78	Whole grain oat	50;100	Usual care	T2DM	FPG; HbA1c; HOMA-IR	1 years; 30 days	Randomized controlled trial
Islam et al., 2015 [33] (Bangladesh)	24 (16/8)	52.83 ± 5.88	Composite flour bread	None	Normal wheat flour	T2DM	FPG;	4 weeks	Randomized controlled trial
Ma et al., 2013 [34] (China)	260 (112/148)	C = 59.30 ± 6.60; I1 = 59.40 ± 6.10; I2 = 60.30 ± 6.00	Organic naked oat with whole germ	50;100	Systematic diet plans	T2DM	FPG; HbA1c; HOMA-IR	30 days	Randomized controlled trial
Lankinen et al., 2011 [35] (Finland)	106 (52/54)	C = 59.00 ± 7.00; I = 58.00 ± 8.00;	Whole grain enriched diet	None	Refined wheat breads	Patients with impaired glucose metabolism	FPG; FPI; HOMA-IR	12 weeks	Randomized controlled trial
Hsu et al., 2008 [36] (Taiwan, China)	11 (6/5)	51.50 ± 16.20	Pre-germinated brown rice	540	White rice	Patients with impaired fasting glucose and type 2 diabetes	FPG; FPI;	12 weeks	Randomized controlled crossover trial
Rave et al., 2007 [37] (Germany)	31 (13/18)	51.00 ± 13.00	Whole grain-based diet- ary product with reduced starch content derived from double-fermented wheat	200	nutrient-dense meal replacement product	Obese subjects with elevated fasting blood glucose	FPG; FPI; HOMA-IR	4 weeks	Randomized controlled crossover trial
Rendell et al., 2005 [38] (USA)	18 (12/6)	62.00 ± 3.00	Prowash barley flakes;	None	A liquid meal replacer	T2DM	G-iAUC; I-iAUC	None	Randomized controlled crossover trial
Jenkins et al., 2002 [39] (Canada)	23 (16/7)	63.00 ± 1.00	Wheat bran	None	Controlled diet with low fiber	T2DM	FPG; HbA1c	12 weeks	Randomized crossover study
Pick et al., 1996 [40] (Canada)	8 (8/0)	46.00 ± 1.00	Oat bran concentrate	None	White bread	Subjects with non-insulin-dependent diabetes	G-iAUC; I-iAUC	24 weeks	Randomized crossover study

**Table 2 nutrients-14-00109-t002:** Subgroup analysis of the effects of whole grain consumption on glycaemic control in diabetic patients.

	Fast Plasma Glucose				Fast Plasma Insulin ^1^			
	No. of Studies	WMD (95% CI)	Heterogeneity		No. of Studies	WMD (95% CI)	Heterogeneity	
			$I^{2}\;(\%)$	*p*			$I^{2}\;(\%)$	*p*
**Study design**								
Parallel	10	−0.68 (−1.03, −0.33)	88.0	<0.001	4	−0.02 (−0.25, 0.21)	42.3	0.158
Crossover	4	−0.39 (−0.79, 0.01)	92.2	<0.001	3	−0.14 (−0.52, 0.24)	56.8	0.099
**Type**								
T2DM	9	−0.84 (−1.29, −0.40)	87.4	<0.001	3	−0.09 (−0.36, 0.18)	52.4	0.122
Others	5	−0.32 (−0.62, −0.02)	92.1	<0.001	4	−0.01S (−0.31, 0.28)	48.1	0.123
**Matching foods**								
Standard diet ^2^	5	−0.22 (−0.44, −0.00)	85.5	<0.001	2	0.28 (−0.12, 0.69)	0	0.464
Others	9	−0.87 (−1.43, −0.31)	90.8	<0.001	5	−0.16 (−0.39, 0.07)	34.7	0.190
**Duration**								
<8 weeks	6	−0.36 (−0.72, 0.00)	86.4	<0.001	2	−0.06 (−0.36, 0.24)	75.2	0.045
≥8 weeks	8	−0.70 (−1.07, −0.33)	91.0	<0.001	5	−0.05 (−0.31, 0.21)	34.2	0.193
**National economic levels**								
Developing	10	−0.84 (−1.20, −0.47)	91	<0.001	3	−0.08 (−0.35, 0.20)	52.7	0.121
Developed	4	−0.03 (−0.11, 0.04)	0	0.857	4	−0.03 (−0.32, 0.26)	48.6	0.120
	**HOMA-IR**				**HbA1c**			
**Study design**								
Parallel	8	−0.34 (−0.70. 0.02)	60.5	0.013	7	−0.74 (−1.19, −0.28)	88.4	<0.001
Crossover	1	−0.90 (−1.95, 0.15)	~	~	2	0.03 (−0.30, 0.36)	24.9	0.249
**Type**								
T2DM	7	−0.42 (−0.80, −0.03)	63.6	0.011	9	−0.58 (−0.88, −0.25)	88.5	<0.001
Others	2	−0.28 (−1.45, 0.90)	63.6	0.097	0	~	~	~
**Matching foods**								
Standard diet ^2^	1	0.30 (−0.65, 1.25)	~	~	1	0.10 (−0.05, 0.25)	~	~
Others	8	−0.46 (−0.83, −0.09)	61.0	0.012	8	−0.70 (−1.11, −0.28)	86.5	<0.001
**Duration**								
<8 weeks	4	−0.24 (−0.43, −0.04)	19.9	0.290	3	−0.26 (−0.54, 0.02)	20.9	0.283
≥8 weeks	5	−1.08 (−2.46, 0.31)	72.2	0.006	6	−0.67 (−1.27, −0.08)	92.6	<0.001
**National economic levels**								
Developing	6	−0.45 (−0.86, −0.03)	69.9	0.006	7	−0.66 (−1.03, −0.30)	91.3	<0.001
Developed	3	−0.24 (−1.02, 0.54)	27.5	0.252	2	−0.16 (−0.53, 0.21)	0	0.552

^1^ Pooled effect of fast plasma insulin was measured by SMD. ^2^ Standard diet means that the matching foods were the same as intervention group without whole grains.

## Data Availability

No new data were created or analyzed in this study. Data sharing is not applicable to this article.

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
