# Peer review of "Role of Whole Grain Consumption in Glycaemic Control of Diabetic Patients: A Systematic Review and Meta-Analysis of Randomized Controlled Trials"

_nutrients, 2021, doi:10.3390/nu14010109_

Round 1

Reviewer 1 Report

The introduction should be implemented by adding  lines describing the profile of nutrients and bioactive compounds (i.e. phenols, lignans, etc)  of whole grains and related properties and related references should be added such as:

Călinoiu, L. F., & Vodnar, D. C. (2018). Whole Grains and Phenolic Acids: A Review on Bioactivity, Functionality, Health Benefits and Bioavailability. Nutrients10(11), 1615. https://doi.org/10.3390/nu10111615

Durazzo et al. Dietary Lignans: Definition, Description and Research Trends in Databases Development. Molecules 201823, 3251. https://doi.org/10.3390/molecules23123251

The description of matter in Lines 61-64 should be enlarged.

The novelty character of this review respect to previous ones should be better marked.

The subparagraph 2.2. Data extraction and study quality should be better explained.

Data in Table 1 should be better described in the text.

Introductory lines should be added to present different subparagraphs of 3.3. Effects of whole grain consumption on parameters involve glycaemic control.

Figures 2-6 should be merged in a single Figure or inserted as Supplementary Material.

The section Conclusion should be greatly implemented by adding limits, advantages and practical applications

Reviewer 2 Report

In the review article titled "Role of whole-grain consumption in glycaemic control of 2 diabetic patients: a systematic review and meta-analysis 3 of randomized controlled trials" the authors analyzed and performed a review of multiple clinical trials that study the effect of whole-grain consumption on glycemic control of the diabetic patient.  The review is on an interesting topic but I have many concerns about the manuscript.

1) Gut microbiota plays an extensive role in the mechanism by which whole grains help lower glucose levels and insulin response. While a few sentences have been written about the gut microbiota the authors need to elaborate and possibly add clinical trials that have looked at gut microbiome changes due to whole grains.

2) In the search strategy only a couple of whole grains have been mentioned by name- wheat, barley (misspelled as barely repeatedly), and oats. Please explain the rationale for not including any other whole grains. As explained in a single line as stated by Tucker et a whole grain sourdough bread could not significantly affect postprandial glucose and insulin in adults with T2DM (Line 358-259)- Similarly, all whole grain foods are unlikely to effectively or positively affect glucose and insulin levels in T2DM patients. This generalization can be misleading.

3) National Economic Levels only look at developed or underdeveloped countries. Could the economic status of the participants of this study be looked at? Participants with lower socioeconomic status may not be able to afford meals with whole grains.

Minor comments-

Figure 7- Overlap of A& B

Line 80- correction - barley

Line 214-215- Needs to be rewritten

Line 292- correction- Pilot

Round 2

Reviewer 2 Report

Strongly suggest adding to the limitations the point that all whole grains and whole grains foods may not be able to positively affect glucose and insulin level and elaborating on it.